# Performance of SARS-CoV-2 serology tests: Are they good enough?

**Isabelle Piec**[1]*, **Emma English**[2], **Mary Annette Thomas**[3], **Samir Dervisevic**[4], **William D. Fraser**[1,5], **William Garry John**[2,5]

1 BioAnalytical Facility, Faculty of Medicine, University of East Anglia, Norwich, United Kingdom, 2 Faculty of Medicine and Health, University of East Anglia, Norwich, United Kingdom, 3 WEQAS, Cardiff and Vale University Health Board, Cardiff, United Kingdom, 4 Virology Department, Norfolk and Norwich University Hospitals, Norwich, United Kingdom, 5 Clinical Biochemistry Department, Norfolk and Norwich University Hospitals, Norwich, United Kingdom

* i.piec@uea.ac.uk

**Data Availability Statement:** All relevant data are within the manuscript and its Supporting information files.

**Funding:** The authors received no specific financial support for this study. They acknowledge the

## Abstract

In the emergency of the SARS-CoV-2 pandemic, great efforts were made to quickly provide serology testing to the medical community however, these methods have been introduced into clinical practice without the complete validation usually required by the regulatory organizations. SARS-CoV-2 patient samples (n = 43) were analyzed alongside pre-pandemic control specimen (n = 50), confirmed respiratory infections (n = 50), inflammatory polyarthritis (n = 22) and positive for thyroid stimulating immunoglobulin (n = 30). Imprecision, diagnostic sensitivity and specificity and concordance were evaluated on IgG serologic assays from EuroImmun, Epitope Diagnostics (EDI), Abbott Diagnostics and DiaSorin and a rapid IgG/IgM test from Healgen. EDI and EuroImmun imprecision was 0.02–14.0% CV. Abbott and DiaSorin imprecision (CV) ranged from 5.2%–8.1% and 8.2%–9.6% respectively. Diagnostic sensitivity of the assays was 100% (CI: 80–100%) for Abbott, EDI and EuroImmun and 95% (CI: 73–100%) for DiaSorin at ≥14 days post PCR. Only the Abbott assay had a diagnostic specificity of 100% (CI: 91–100%). EuroImmun cross-reacted in 3 non-SARS-CoV-2 respiratory infections and 2 controls. The DiaSorin displayed more false negative results and cross-reacted in six cases across all conditions tested. EDI had one cross-reactive sample. The Healgen rapid test showed excellent sensitivity and specificity. Overall, concordance of the assays ranged from 76.1% to 97.9%. Serological tests for SARS-CoV-2 showed good analytical performance. The head-to-head analysis of samples revealed differences in results that may be linked to the use of nucleocapsid or spike proteins. The point of care device tested demonstrated adequate performance for antibody detection.

## Introduction

The scientific community has had to rapidly develop and manufacture tests for the new SARS-CoV-2 pandemic at unprecedented speed, taking three months to develop assays that would ordinarily take three years. Serology testing, that can identify those who have previously

Norfolk, Suffolk, Essex and Bedfordshire Freemasons for their generous material support in providing funding for some equipment used in this study.

**Competing interests:** The authors acknowledge the Norfolk, Suffolk, Essex and Bedfordshire Freemasons for their material support in providing funding for some equipment used in this study. This does not alter our adherence to PLOS ONE policies on sharing data and materials.

**Abbreviations:** anti-CCP, Cyclic citrullinated peptide antibodies; CLSI, Clinical and Laboratory Standards Institute; CR, Cross-reactivity; CV, Coefficient of variation expressed as percentage; EBV, Epstein Barr Virus; EDI, Epitope Diagnostics Ltd; EQA, External quality assessment; IgG, Immunoglobulin G; LFIAs, Lateral flow immunoassays; N, Negative control; NNUH, Norwich and Norfolk University Hospital; NOAR, Norfolk Arthritis Register; OD, Optical density; OD, Optical density/absorbance; P, SARS-CoV-2 Positive; PHE, Public Health England; POCT, Point of Care Testing; QEH, Queen Elizabeth Hospital in King Lynn; R, Threshold of positivity; RA, Rheumatoid Arthritis; RLU, Relative Light Unit; S1/ S2, Spike protein 1 and 2; TSI, Thyroid stimulating immunoglobulin; WEQAS, Wales External Quality Assessment Scheme (UK).

been exposed to the SARS-CoV-2 virus and have mounted an immune response, has been hailed as key to managing the pandemic however controversy remains over both the accuracy and utility of serology testing in disease management. Structural proteins, including the spike (essential for viral infection) and the nucleocapsid (important for viral RNA transcription), are both potential targets for early detection of infection and known to elicit an immune response in the host [1] with antibodies detectable within 20 days of disease onset [2–4].

Systematic reviews [5, 6] challenged the diagnostic accuracy of serological tests, particularly when using lateral flow immunoassays (LFIAs). Public Health England (PHE) showed only the Siemens and the Roche Diagnostics assays met the minimum UK Medicines and Health-care products Regulatory Agency Target Product Profile criteria for sensitivity [7] after the threshold of positivity was adjusted to 0.128. Assays from DiaSorin and Abbott Diagnostics [8] also provided acceptable diagnostic results. These evaluations did not address cross-reactivity. To our knowledge little has been done regarding interference from antibodies produced dur-ing other viral infection and autoimmune disorders. Additionally, with the focus on diagnostic sensitivity and specificity, little has been done to evaluate the analytical accuracy, which if poor, has the potential to negate all of these study findings. Indeed in the editorial, Duong and colleagues clearly states that there is a need for critical independent evaluations of these tests, using the same specimen panels [9].

This study provides a head-to-head evaluation of the diagnostic and analytical performance of four commercially available IgG based serology assays for SARS-CoV-2 and a diagnostic accuracy study of one point of care LFIA.

## Materials and methods

### Specimen collection and storage

Patients were not involved in any part of the work. All samples were from archived specimens and were fully anonymized before we accessed them. Therefore, our study is in accordance with the blanket Ethical standards of University of East Anglia on de-identified samples for method development. Moreover, using the UK NHS Research Ethics Committee decision toolkit (http://www.hra-decisiontools.org.uk/ethics/) we confirmed that separate ethical review is not required for this study which is in concordance with the Helsinki Declaration.

All serum samples were collected between April and June 2020, anonymized, aliquoted and stored at –80°C until analyzed. SARS-CoV-2 PCR-positive patients (AusDiagnostics platform, Chesham, UK) were of both genders, age range 66 to 93 and hospitalized at the Norfolk and Norwich University Hospital (NNUH) or Queen Elizabeth Hospital in King Lynn (QEH). Samples were taken 8–44 days after testing positive for SARS-CoV-2. Negative control samples were collected in 2018 from patients with no history of infection or immune disorder. Pre-pandemic samples from patients who had a range of confirmed respiratory infections (includ-ing Influenza A, B and seasonal coronaviruses [Table 1]), samples collected from patients with inflammatory polyarthritis positive for anti-cyclic citrullinated peptide antibodies (anti-CCP) along with samples positive for thyroid stimulating immunoglobulin (TSI) were used to test the non-specific binding of non-SARS-CoV-2 antibodies. These groups of samples are referred to as N (negative control), CR (cross-reactivity), RA (Rheumatoid Arthritis), TSI (patients with thyroid stimulating immunoglobulin) and P (SARS-CoV-2 Positive). A total of 195 indi-vidual serum samples (43 P, 50 N, 50 CR, 22 RA and 30 TSI) were analyzed for SARS-CoV-2 IgG antibodies. For a subset of patients, samples were available for a series of time-points thus allowing for a time course analysis (43 patients, 142 samples).

**Table 1. Respiratory infections tested for cross reactivity in the SARS-CoV-2 IgG immunoassays.**

| Infection | No patients |
|---|---|
| Epstein-Barr virus | 8 |
| Influenza A virus | 8 |
| Respiratory syncytial virus | 7 |
| Seasonal Coronaviruses | 7 |
| Borrelia burgerdorferii | 4 |
| Cytomegalovirus | 3 |
| Varicellazoster virus | 3 |
| Bordella Pertussis | 2 |
| Hepatitis B | 2 |
| Human immunodeficiency virus | 2 |
| Adenovirus | 1 |
| Mycoplasma | 1 |
| ParaInfluenza | 1 |
| Rhinovirus | 1 |

## Study design

SARS-CoV-2 IgG immunoassays were from 1) Epitope Diagnostics Inc. (EDI, San Diego, CA, USA) performed using the Agility ELISA automate (Dynex Technologies, Chantilly, VA, USA), 2) EuroImmun UK ITC (UK) performed manually, 3) Abbott Diagnostics (Maidenhead, UK) on the Alinity™ i analyzer and 4) DiaSorin (London, UK) on the Liaison XL analyzer. A subset of samples was also tested using the point of care testing (POCT) device SARS-CoV-2 IgG/IgM rapid test from Healgen (Houston, TX, USA). Due to a limited number of cassettes available, 49 samples from 27 P were analyzed along with 3 N, 8 CR, 4 RA and 4 TSI. Cross-reactive and negative samples were primarily chosen from patient samples proven positive for seasonal coronaviruses and influenza A or a false positive result in one or more of the immunoassays. We focused on the IgG results in order to compare with the immunoassays.

Assays were performed by trained biomedical scientists using manufacturer's instructions. The SARS-CoV-2 Abbott assay was performed in the clinical biochemistry department at NNUH and the other SARS-CoV-2 assays were performed at the University of East Anglia. All other non- SARS-CoV-2 related tests were performed at the NNUH virology department. DiaSorin SARS-Cov-2 is a quantitative assay and antibody concentrations are expressed in AU/mL. The Abbott, EDI and EuroImmun are qualitative assays for which the result is calculated using the ratio of the sample optical density (OD) against the negative or calibrator control (S1 Table). EuroImmun and DiaSorin assays detect antibodies to, respectively, recombinant S1 and S1/S2 domains of the SARS-CoV-2 spike protein while both the EDI and Abbott assay detect antibodies to the nucleocapsid. The POCT from Healgen is a solid phase lateral flow immunochromatographic assay (LFIA) for detection of SARS-CoV-2 of IgG and IgM, antigen not specified.

## Imprecision

As the results are expressed with a values correlating with the amount of antibody detectable, imprecision was assessed using a Clinical and Laboratory Standards Institute (CLSI) EP-15 based protocol on the automated clinical laboratory analyzers protocol [10]. Positive and

negative patient pools and/or controls of different concentrations were prepared and frozen as aliquots and assayed as 5 replicates per day on 5 different days. For the plate-based assays, inter- and intra-assay CVs were calculated. Intra-assay was determined using the CV of the optical density (OD) of duplicated samples. Inter-assay was determined using the CV obtained from the sample pool and the kit positive control across the plates.

## Statistics

Using IBM SPSS Statistics 25.0.0.1, Mann-Whitney and Cohen's Kappa tests were used to compare OD results between groups and to determine the concordance between the assays, respectively. Analysis of EP15 was performed using EP evaluator. Variation was estimated on calculated values (R) or response (Relative Light Unit, RLU) as intra and inter-assay coefficient of variation (CV). Graphical representations were conducted with GraphPad Prism version 8.0 (GraphPad Software, Inc., USA). Throughout the tables, figures, and legends, the following terminology is used to show statistical significance: $^{*}P<0.05$; $^{**}P<0.01$ and $^{***}P<0.001$.

## Results

### Imprecision

**Abbott.** EP15 and was performed on two Alinity analyzers (Table 2). Overall, negative pool imprecision was CV = 8.1% and 6.8% on equipment 1 and 2 respectively. Positive pool imprecisions were CV = 2.3% and 1.1% respectively.

**DiaSorin.** EP15 imprecision was estimated based on response intensity (RLU). Positive control imprecision was between 8.2% and 13.8% (Table 2). The negative quality control material results were consistently below the lower limit of detection of 3.8AU/mL and the negative

**Table 2. EP15 analysis on two Abbott Alinity, DiaSorin Liaison XL and ELISAs imprecision tests.**

| | Sample | n | Mean | Intra-assay imprecision | | Inter-assay imprecision | |
|---|---|---|---|---|---|---|---|
| | | | | SD | %CV | SD | %CV |
| **ABBOTT** | Alinity 1 (Neg) | 25 | 0.136 (R) | 0.011 | 8.1 | 0.011 | 8.1 |
| | Alinity 1 (Pos) | 25 | 7.254 (R) | 0.167 | 2.3 | 0.170 | 2.3 |
| | Alinity 2 (Neg) | 25 | 0.143 (R) | 0.007 | 5.2 | 0.010 | 6.8 |
| | Alinity 2 (Pos) | 25 | 7.242 (R) | 0.081 | 1.1 | 0.082 | 1.1 |
| **DIASORIN** | Kit Negative control | 20 | 2457 (RLU) | 1730 | 70.4 | 2860 | 116.4 |
| | Level 1 (Neg pool) | 25 | 6945 (RLU) | 4003 | 57.6 | 5887 | 84.8 |
| | Kit Positive control | 25 | 58662 (RLU) | 4815 | 8.2 | 5608 | 9.6 |
| | Level 2 (Pool 1) | 25 | 83236 (RLU) | 11128 | 13.4 | 11128 | 13.4 |
| | Level 3 (Pool 2) | 25 | 410600 (RLU) | 56802 | 13.8 | 56802 | 13.8 |
| | Level 4 (pool 3) | 25 | 557660 (RLU) | 55667 | 10.0 | 58092 | 10.4 |
| **EDI** | Kit Negative control | 27 | 0.074 (OD) | - | - | 0.014 | 10.9 |
| | Kit positive control | 9 | 0.482 (OD) | - | - | 0.068 | 14.2 |
| | Duplicate samples | 308 | - | 3.8 | 3.3 | - | - |
| **EURO – IMMUN** | Kit Negative control | 3 | 0.074 (OD) | - | - | 0.003 | 3.7 |
| | Kit positive control | 3 | 1.169 (OD) | - | - | 0.15 | 2.9 |
| | Calibrator | 3 | 0.277 (OD) | - | - | 0.027 | 9.5 |
| | Duplicate samples | 44 | - | 6.7 | 6.1 | - | - |

For the DiaSorin, negative samples (QC or pools) results were typically below the limit of detection of 3.8 AU/mL and variation was estimated on the response in relative light units (RLU).

pool concentration was consistently below 10AU/mL, the resulting calculated imprecision was therefore expectedly elevated.

**EDI and EuroImmun.** Intra-assay imprecision on duplicate samples (Table 2) was on average CV = 3.3±3.8% and 6.1±6.7% respectively. Inter-assay imprecision of EDI was CV = 14.2% for the kit positive pool and 16.5% for the negative pool. Baseline OD varied between the plates increasing the inter-assay variations, however, the ratio positive/cut-off was on average 1.43 ±0.16 (CV = 11.1%). Inter-assay of EuroImmun was evaluated using the positive kit QC, the calibrator and the negative kit control. Coefficient of variation were CV = 12.9%, 9.5% and 3.7% respectively.

## Specificity and sensitivity

A total of 43 individual P was analyzed for SARS-CoV-2 IgG antibodies. Of these, twenty had samples taken at least 14 days after a positive PCR result (P≥14) and 23 were taken prior (P<14). All P≥14 had detectable antibodies in the EDI, EuroImmun, Abbott and Healgen assays. However, one sample returned a negative result using the DiaSorin assay. These results suggest a true positive rate of 100% with EDI, EuroImmun, Abbott and Healgen assays and 95% for the DiaSorin assay.

Amongst the 23 P<14 samples, antibodies were detected for 65% (Abbott & EuroImmun), 61% (EDI) and 43% (DiaSorin) of the samples. Two samples R were close to the threshold in EDI and Abbott (EDI: 0.8 and Abbott 1.9; EDI: 1.0 and Abbott 0.8) resulting in one being positive in one assay and negative in the other (and vice-versa).

All 50 N were negative on the Abbott and EDI. Two samples were positive and 48 were negative on the EuroImmun (although 2 were equivocal). Two false positive samples were also observed on the DiaSorin, one being positive on both DiaSorin and EuroImmun assays.

The IgG kits showed a very good diagnostic ability to differentiate between P and N (Table 3). Overall, EuroImmun and DiaSorin showed lower sensitivity and specificity than EDI and Abbott. Sensitivity ranged between 81–100% on all time points for EDI, EuroImmun and Abbott. DiaSorin sensitivity was 71% on all time points and 95% for P≥14. Specificity was consistently 100% for the Abbott while it ranged between 92 to 100% for the other assays.

## Cross-reactivity

There were no SARS-CoV-2 IgG positive results from patients with non-SARS-CoV-2 infection (CR, n = 50, including seasonal flu (n = 7)), anti-CCP positive (RA, n = 22) nor TSI positive (n = 30) using the Abbott and the EDI assays. Overall, DiaSorin showed the highest (4%) cross-reactivity (2CR, 1 RA and 1 TSI), followed by EuroImmun (3%–3CR) and EDI (1%–1 TSI). The Mann-Whitney test showed that on the EDI only, the R value of samples used to test cross-reactivity (RA and TSI) was significantly elevated, however only one sample was falsely positive for SARS-CoV-2 (Fig 1).

Any sample that gave a false positive result in any of the immunoassays was also tested on the Healgen POCT and none were IgG positive. However, a very weak signal could be detected on one TSI sample and one sample from a patient with seasonal flu. Because of the very small number of samples tested; specificity calculation was not performed for the rapid test.

## Time course analysis

We analyzed 1 to 13 data points for 43 P. We observed an increase of the signal for presence of IgG over time going from negativity to positivity and reaching a plateau (Fig 2). Sigmoid curve-fitting indicated a time from PCR to seroconversion at 9.8 days (95% CI 10.7–13.7), 10.2 (95% CI 8.5–11.8), 12.2 days (95% CI 10.7–13.7) and 10.4 days (95% CI 7.9–12.9) for EDI,

**Table 3. Sensitivity of the assays was estimated on all time points and including only samples >14 days post PCR.**

| | | Assay | Total Tested | SARS-CoV-2 IgG Positive | SARS-CoV-2 IgG Negative | Equivocal result | Result (95% CI) |
|---|---|---|---|---|---|---|---|
| SENSITIVITY | **SARS-CoV-2 Positive all time points** | EDI | 43 | 35 | 8 | n/a | 81 (66–91) |
| | | EuroImmun | 43 | 35 | 8 | 0 | 81 (66–91) |
| | | Abbott | 43 | 35 | 8 | n/a | 81 (66–91) |
| | | DiaSorin | 42 | 30 | 12 | 0 | 71 (55–84) |
| | | Healgen | 27 | 27 | 0 | n/a | 100 (84–100) |
| | **SARS-CoV-2 Positive ≥14 days post PCR** | EDI | 20 | 20 | 0 | n/a | 100 (80–100) |
| | | EuroImmun | 20 | 20 | 0 | 0 | 100 (80–100) |
| | | Abbott | 20 | 20 | 0 | n/a | 100 (80–100) |
| | | DiaSorin | 20 | 19 | 1 | 0 | 95 (73–100) |
| | | Healgen | 20 | 20 | 0 | n/a | 100 (80–100) |
| SPECIFICITY | **Pre-pandemic controls (N)** | EDI | 50 | 0 | 50 | n/a | 100 (91–100) |
| | | EuroImmun | 50 | 2 | 46 | 2 | 92 (79–97) |
| | | Abbott | 50 | 0 | 50 | n/a | 100 (91–100) |
| | | DiaSorin | 50 | 2 | 48 | 0 | 96 (85–99) |
| | | Healgen | 4 | 0 | 4 | n/a | - |
| | **Other Respiratory Infection (CR)** | EDI | 50 | 0 | 50 | n/a | 100 (91–100) |
| | | EuroImmun | 50 | 3 | 47 | 0 | 94 (82–98) |
| | | Abbott | 50 | 0 | 50 | n/a | 100 (91–100) |
| | | DiaSorin | 50 | 2 | 48 | 0 | 96 (85–99) |
| | | Healgen | 9 | 1 | 8 | n/a | - |
| | **Rheumatoid Arthritis (RA)** | EDI | 22 | 0 | 22 | n/a | 92 (72–99) |
| | | EuroImmun | 22 | 0 | 22 | 0 | 92 (72–99) |
| | | Abbott | 22 | 0 | 22 | n/a | 100 (82–100) |
| | | DiaSorin | 22 | 1 | 21 | 0 | 95 (75–100) |
| | | Healgen | 4 | 0 | 4 | n/a | - |
| | **Thyroid Disorder (TSI)** | EDI | 30 | 1 | 29 | n/a | 97 (81–100) |
| | | EuroImmun | 30 | 0 | 28 | 2 | 93 (76–99) |
| | | Abbott | 30 | 0 | 30 | n/a | 100 (85–100) |
| | | DiaSorin | 30 | 1 | 29 | 0 | 97 (81–100) |
| | | Healgen | 4 | 1 | 3 | n/a | - |

Specificity was estimated on pre-2020 samples (N) from healthy individuals and patients with disorders that induce the production of potentially interfering substances.
n/a = no equivocal range available.

Abbott, DiaSorin and EuroImmun assays respectively. Note that due to a limited number of EuroImmun tests available, we only had measurements for 56 (of 142) data points. One data point was missing for Abbott and 4 were missing for DiaSorin due to insufficient sample volume.

We tested 48 samples from 27 P patients using the Healgen rapid test. Ninety four percent (n = 45) displayed a positive test for IgG. Samples showed positive results with POCT from day 7 post PCR although these were still negative in the other immunoassays (SARS-CoV-2 positive at day 12).

## Assay concordance

Abbott and EDI had the greatest concordance with Cohen's Kappa of 0.957 and 97.9% agreement between the all results (Table 4). DiaSorin was the most different, with agreements below

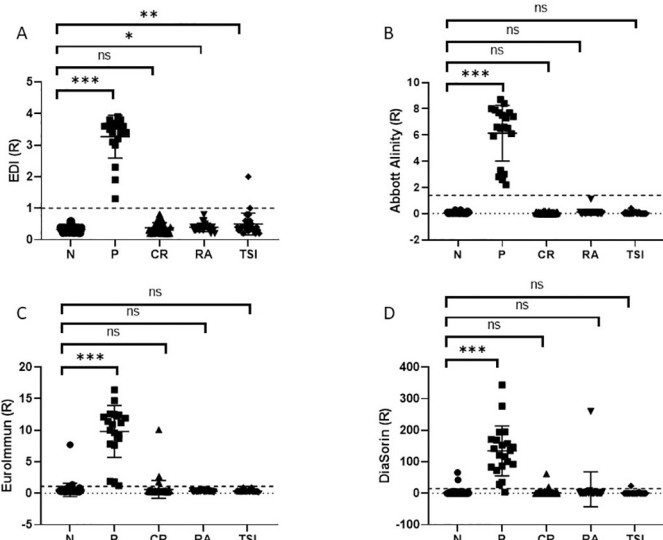

**Fig 1. Dot-plots of R values for each condition (N, P, CR, RA and TSI) for the (A) EDI, (B) Abbott, (C) EuroImmun and (D) DiaSorin tests.** Mann-Whitney analysis demonstrated a significant increase in the R value for the positive samples. Mann-Whitney statistical significance $^*p<0.05$; $^{**}p<0.01$ and $^{***}p<0.001$. Dotted line represents the positive cut-off for each assay.

95%. The Healgen POCT concordance with the other assays was low (below 90%) but reflect a limited number of samples and may not be representative. Modifying the threshold to 0.8 for EDI would allow the detection of 2 more P<14 without increasing the rate of false positive. No change in threshold in the other assay would reclassify any results without dramatically affecting the specificity to either have a high rate of false positive or false negative.

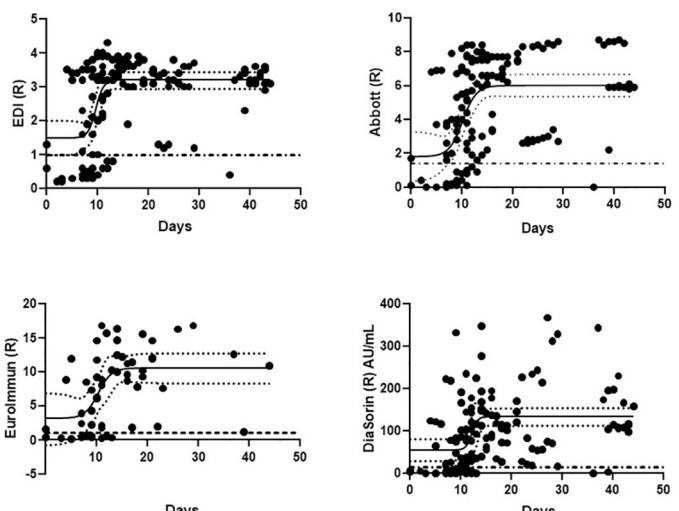

**Fig 2. Seropositivity in specimen with PCR positive relative to day of PCR.** Dashed line represents the cut-off ratio for each assay. Solid black line and dotted lines represent the 4-parameter logistic curve-fit of the points with confidence interval. Time to PCR onset is calculated as curve inflection point.

**Table 4. Cohen's Kappa concordance analysis of the assays and overall (all samples included) agreement of results given as %.**

| Abbott | Euroimmun | DiaSorin | Healgen | Cohen's Kappa (±SD) % agreement | |
|---|---|---|---|---|---|
| 0.950 (0.019) 97.9% | 0.907 (0.034) 96.6% | 0.892 (0.027) 94.8% | 0.745 (0.083) 88.2% | EDI | |
| | 0.892 (0.037) 96.1% | 0.891 (0.027) 94.8% | 0.777 (0.078) 89.6% | Abbott | |
| | | 0.852 (0.043) 94.7% | 0.631 (0.111) 82.7% | EuroImmun | |
| | | | 0.485 (0.109) 76.1% | DiaSorin | |

Equivocal results were considered negative.

## Discussion

### Statement of principal findings

In this head to head study we demonstrated the good performance of four commercially available serologic assays for SARS-CoV-2 and one POCT. Abbott, Epitope Diagnostics Ltd and EuroImmun demonstrated higher sensitivity and specificity than the DiaSorin assay on the same specimens. The Abbott assay showed no cross-reactivity to any other potential interfering substances tested while EDI, EuroImmun and DiaSorin cross-reacted in 1%, 3% and 4% of the sample tested. However, no assay cross-reacted with Influenza A and B or other coronaviruses. The analytical performance was deemed acceptable although it varied considerably between the different methods.

It is estimated that there are nearly 300 different SARS-CoV-2 antibody tests in development globally ranging from POCT through to assays on large clinical laboratory analyzers. Whilst data is accruing on the sensitivity and specificity of a number of these assays [5, 6] there are still many with little or no published, independent performance evaluations. Whilst there is a focus on the diagnostic accuracy of these tests, much less is understood about the analytical performance of these devices such as imprecision and cross reactivity with common respiratory illnesses or immunoassay interferences. Without this knowledge the sensitivity and specificity data is brought into question and it is important that the limitations of assay are fully understood before applying the results in clinical practice. The Food and Drug Administration and European Medicines Agency acceptance criteria for biological assays typically define the required between-run and within-run precision as CV$\leq$15% for positive samples and $\leq$20% for samples at the lower limit of quantification [11, 12]. All immunoassays passed the criteria for positive samples.

Published median seroconversion time for IgG is around 14 days post symptoms [13–15]. As we did not have access to symptom onset for most patients, we used PCR day to date the samples, before and after day 14. We included in the positive group only one sample per patient, thus limiting our sample size. However, our results are not biased by repeat measurements. All samples post day 14 were positive in all assay except DiaSorin, which returned one false negative (day 39). Positivity prior to day 14 was consistent between EDI, EuroImmun and Abbott. These results are differing from those published by PHE who observed more false negative results in the Abbott than the DiaSorin (92.7% sensitivity vs 95% sensitivity, respectively) [8]. We estimated seroconversion post PCR positivity to be between 9 and 12 days on

these assays. Although we couldn't do a full comparison of the POCT with the immunoassays, 100% of the P≥14 samples were IgG positive. More samples were also positive with POCT prior day 14 than in the other assays.

In regard to the POCT, our study showed excellent sensitivity and specificity. We observed no false negative results on P≥14 after a positive SARS-CoV-2 PCR and more samples were IgG positive P<14 than the other immunoassays. Two potential false positive were detected (including seasonal flu) but the signal was very weak and confirmation would be necessary. The results of systematic reviews on point-of care serological tests for SARS-CoV-2 suggest discontinuing the use of the devices due to low sensitivity [5]. Our results tend to reveal a different pattern however we only performed a limited number of tests.

We analyzed 50 samples collected in 2018 from patients with no known infection as negative controls. Both the EDI and the Abbott showed 100% specificity. However, EuroImmun and DiaSorin produced false positives (n = 4 and 2, respectively). Only one of these samples was common between both assays. PHE also showed lower specificity of the DiaSorin assay (vs Abbott). We analyzed 50 samples from patients (pre-pandemic) presenting with respiratory infection. Among those 7 had the seasonal flu, 8 had influenza A., other viruses included EBV, Varicellazoster virus, parainfluenza, Adenovirus. EDI and Abbott showed 100% specificity with no false positive; however, we observed 3 positive results with the EuroImmun, two of these also being positive with the DiaSorin. These samples were from patients with EBV (n = 1) and RSV (n = 2). Our results on EuroImmun differ slightly from a previous evaluation [16], where specificity of the assay was excellent as early as 4 days after positive PCR and only 2 of 28 samples showed borderline cross-reactivity to common human coronaviruses. None of the assays showed cross-reactivity either to the seasonal CoV flu or to Influenza A. Although it is based on a small number of sample (n = 7 for each), it brings confidence that assays will be able to discriminate SARS-CoV-2 antibodies during the next seasonal flu. Tang *et al.*, showed similar results on 5 patients using EuroImmun and Abbott Assay [17]. A great variety of endogenous substances such as polyreactive antibodies or autoantibodies, can interfere with the reaction between analyte and reagent antibodies in immunoassays. Assays for SARS-CoV-2 are no exception. Manufacturers, and evaluation studies to date, offer a limited insight into cross-reactivity of other antibodies in particular to other SARS-CoV antibodies [18–22]. A small independent study showed no cross-reactivity was seen for patients with Influenza A (n = 2), Influenza B (n = 2) and other coronaviruses (n = 5) [17]. Samples with potentially interfering antibodies did not cross-react in the Abbott Diagnostics assay, and a limited number cross-reacted in the other assays. Using the EDI assay, the signal obtained for both RA and TSI samples is significantly higher than the negative controls however all but one TSI sample remain below the cut-off of positivity. The cut-off is therefore appropriate for use with the assay with potentially cross-reactive substances. None of these samples was common between the different assays and modification of the various threshold would not improve performance of any assay.

Successful attempts to treat SARS-CoV-2 patients with blood from convalescent individuals suggest antibodies against SARS-CoV-2 may have the ability to confer protective immunity to the disease [23–28]. Spike proteins are the most likely target for neutralizing antibodies are displayed on the surface of the virus whereas the nucleocapsid is contained within the viral envelope [29, 30]. Antibodies against the nucleocapsid have been shown to appear first [31, 32], followed by the production of antibodies against the spike protein [13, 14]. Therefore, assays based on the nucleocapsid detection appear to be more sensitive early on in the disease recovery but presence of anti-S1/S2 antibodies may indicate presence of neutralizing antibodies. Both the EuroImmun and the DiaSorin are targeted the spike protein of SARS-Cov-2 while the EDI and Abbott are targeted to the nucleocapsid protein of the virus. We observe a highest

specificity of both nucleocapsid assays (EDI and Abbott, 100% (91–100%)) compared to the two spike assays (DiaSorin (96% (85–99%)) and EuroImmun (92% (79–97%))). Although the EuroImmun assay had the same sensitivity (all time points to PCR) as the EDI and Abbott, the DiaSorin assay was less sensitive (71% (73–100%) vs (81% (66–91%))), potentially supporting this hypothesis.

Overall, the assays had high concordance, DiaSorin being the least identical to the others, with higher false negative and false positive, and lower performance. This is in accordance with the high false positive rate observed by Boukli et al. [33] with the DiaSorin Liaison SARS-CoV-2 IgG assay on patients with non-SARS-CoV-2 acute infections. In April 2020 both the DiaSorin assay and the Abbott assays were authorized by Public Health England for emergency use in the clinical setting; 15 sites using the Abbott method reported to the WEQAS scheme in October while 84 (Abbott Architect and Alinity) reported to the UKNEQAS scheme in November. However, 3 sites using the DiaSorin method reported to WEQAS and 9 to UKNE-QAS for the same period. The same samples were analyzed on the different platforms and therefore the direct comparison is possible. However, one needs to consider the potential variance in antigen as the Wuhan strain has evolved as geographic spread has occurred between the different regions of the globe (GISAID) [34] and it is possible that these differences will not be seen on a different set of samples. Harmonization of the assays is necessary but will be near impossible with such variation between assay designs (spike vs nucleocapsid). The Wales External Quality Assessment Scheme (WEQAS, UK, https://www.weqas.com/) and the UK NEQAS (https://ukneqas.org.uk/) are now offering a SARS-CoV-2 antibody external quality assessment (EQA) program for laboratories which will reduce uncertainty associated with different methods.

## Strengths and limitations of this study

The main strength of our study is the direct comparison (same specimens) of five SARS-CoV-2 assays and the analysis of potentially cross-reactive substances produced during other respiratory infections and disorders such as rheumatoid arthritis and thyroid imbalance which are known to affect immunoassays.

Limitations include the limited number of positive samples due to the UK East Anglian region's low prevalence and unavailability of onset date of SARS-CoV-2 symptoms. PCR may have been done from symptom onset day to several days post symptom; therefore, we based our seroconversion on PCR-positive date. The severity of symptoms was not available for all patients however these patients were hospitalized and we cannot comment on whether the production of antibodies correlate with the severity of symptoms.

## Conclusion and policy implications

The role of serology testing in the management of people with SARS-CoV-2 infection will remain controversial until we have clear data that enables an understanding of how production of IgG relates to immunity over time and whether or not the presence or absence of antibodies can inform risk of future infection. Whilst the clinical utility of serology tested is debated, it is important that the diagnostic and analytical performance of these tests is understood and adequate for need so that there can be confidence in the results when a meaningful clinical use is determined. Without high quality analytical testing the clinical application of serology testing in the future is not viable.

This study examines the performance of four commercially available serologic assays for SARS-CoV-2 in a head to head study. Our study demonstrated good analytical performance for all of the assays, however we observed Abbott, EDI and EuroImmun demonstrated higher

sensitivity and specificity than the DiaSorin assay in this study. Whilst a full evaluation was not possible the P14+ samples from the main study were used in a sub analysis using the Healgen POCT device which showed 100% specificity, this contradicts earlier studies [5, 6] and indicates that the evolution of the quality of POC devices has been rapid and some may now demonstrate adequate performance for antibody detection.

Assays showed 0–4% cross-reactivity, however none with Influenza viruses. This may give increase confidence of the test during the seasonal flu period. We observed differences between the assay responses with DiaSorin being the most different from the other three. We hypothesize that these differences may be linked to the design of the assay themselves (spike glycoprotein or nucleocapsid) and the timeline of production of antibodies for either antigen. We also suggested the possibility that the antigen plasticity and the antigen used when the manufacturer set up the test may influence the sensitivity of the CoV-2 assays. These findings highlight the importance of following the evolution of the antibody production and evolution of the virus over time. But it also highlights how harmonization of the assays will be complex.

## Supporting information

**S1 Table. Characteristics of the immunoassays evaluated, as provided by the manufacturers.**
(DOCX)

## Acknowledgments

The authors would like to thank the staff at the Norfolk and Norwich University Hospital and the Queen Elizabeth Hospital, Kings Lynn who collected the samples from SARS-CoV-2 patients, in particular Christopher Jeanes, and the Norfolk Arthritis Register (NOAR) for kindly providing historical samples of patients with inflammatory polyarthritis. We would also like to express our gratitude to Myra Del Rosario and Simon Clements who analyzed the samples on the Abbott Alinity, and Christopher McDonnell and Reenesh Prakash who procured EuroImmun kits.

## Author Contributions

**Conceptualization:** Isabelle Piec, Emma English, Samir Dervisevic, William D. Fraser, William Garry John.

**Data curation:** Isabelle Piec.

**Formal analysis:** Isabelle Piec.

**Funding acquisition:** Samir Dervisevic, William Garry John.

**Investigation:** Isabelle Piec.

**Methodology:** Isabelle Piec, Emma English, William D. Fraser, William Garry John.

**Project administration:** Isabelle Piec, Emma English, Samir Dervisevic.

**Resources:** Isabelle Piec, Mary Annette Thomas, Samir Dervisevic, William Garry John.

**Supervision:** William Garry John.

**Validation:** Isabelle Piec.

**Writing – original draft:** Isabelle Piec.

**Writing – review & editing:** Emma English, Mary Annette Thomas, Samir Dervisevic, William D. Fraser, William Garry John.

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
