## [Decision Letter · Decision Letter 0]

9 Dec 2020

PONE-D-20-31021

Performance of SARS-CoV-2 Serology tests: Are they good enough?

PLOS ONE

Dear Dr. Piec,

Thank you for submitting your manuscript to PLOS ONE. After careful consideration, we feel that it has merit but does not fully meet PLOS ONE’s publication criteria as it currently stands. Therefore, we invite you to submit a revised version of the manuscript that addresses the points raised during the review process.

ACADEMIC EDITOR: Please review comments made by the reviewers and provide point by point response in your revised manuscript.

We look forward to receiving your revised manuscript.

Kind regards,

Muhammad Adrish

Academic Editor

PLOS ONE

Journal Requirements:

2)  Thank you for stating the following in the Acknowledgments Section of your manuscript:

[We finally wish to thank the Norfolk, Suffolk, Essex and Hertfordshire Freemasons for their generous financial support to purchase the Agility system.]

 [The funders had no role in study design, data collection and analysis, decision to

publish, or preparation of the manuscript.]

3) Please include captions for your Supporting Information files at the end of your manuscript, and update any in-text citations to match accordingly. Please see our Supporting Information guidelines for more information: http://journals.plos.org/plosone/s/supporting-information.

Reviewers' comments:

Reviewer's Responses to Questions

**Comments to the Author**

1. Is the manuscript technically sound, and do the data support the conclusions?

Reviewer #1: Yes

Reviewer #2: Yes

Reviewer #3: Yes

2. Has the statistical analysis been performed appropriately and rigorously? 

Reviewer #1: Yes

Reviewer #2: Yes

Reviewer #3: Yes

3. Have the authors made all data underlying the findings in their manuscript fully available?

Reviewer #1: Yes

Reviewer #2: Yes

Reviewer #3: Yes

4. Is the manuscript presented in an intelligible fashion and written in standard English?

Reviewer #1: Yes

Reviewer #2: Yes

Reviewer #3: Yes

5. Review Comments to the Author

Reviewer #1: Major comment: The manuscript# PONE-D-20-31021entitled “Performance of SARS-CoV-2 Serology tests: Are they good enough?” assess the performance of SARS-CoV-2 serologic tests available in clinical practices.

These methods were introduced into clinical practice quickly without an extensive validation usually required by the regulatory authorities. The IgG serologic Assays from EuroImmun, Epitope Diagnostics (EDI), Abbott Diagnostics and DiaSorin and a rapid IgG/IgM test from Healgen were evaluated for the imprecision, diagnostic sensitivity and specificity and concordance. The SARS-CoV-2 patient samples were analyzed as pre-pandemic control specimen with confirmed respiratory infections, inflammatory polyarthritis and several samples positive for thyroid stimulating immunoglobulin and post pandemic samples. Authors showed that the diagnostic sensitivity of the assays were 100% (CI: 80-100%) for Abbott, EDI and EuroImmun and 95% (CI: 73-100%) for DiaSorin at ≥14 days post infection. Only the Abbott assay had a diagnostic specificity of 100% (CI: 91-100%). The DiaSorin displayed more false negative results across all conditions tested. The imprecision for EDI and EuroImmun, was 0.02-14.0% CV while for Abbott and DiaSorin imprecision (CV) ranged from 5.2% - 8.1% and 8.2% - 9.6% respectively. Overall, concordance of the assays ranged from 76.1% to 97.9%. The point of care rapid test Healgen, showed excellent sensitivity and specificity. Authors attributed the differences in results to the use of nucleocapsid and spike proteins. The point of care device tested demonstrated adequate performance for antibody detection.

Limited sample size should be included in the limitation of the study.

The Figure legend describes Figure 1 as Box-plot even though data shown as a dot plot.

The Figure 1A reveled very little to no cross reactivity. The statistical analysis among groups show significant differences between N to RA and N to TSI. None of the values in RA and only one value in TSI groups is above the positive cutoff value. May be author should comment on meaning of such statistical significance.

In this study the DiaSorin displayed the most-false negatives and showed cross-reactivity when comparing with other serologic assays. It will important to add in the discussion the frequency of use of these tests within UK or globally specially if that data is available through health care systems or through the company.

Reviewer #2: This is a well written manuscript. COVID -19 is also a huge demanding topic. I have some minor comments.

Overall: The number of positive samples is very low (43). Though the present analysis showed a good agreement. But it may alter if the sample size increased. There are so many published paper with the information of different type of serological testing of SARS-CoV-2. Presently we need more specific decision for specific serological test.

Please add some explanation in discussion section, why the specificity of DiaSorin assay showed low.

Line 111: Please add the time period of the SARS-CoV-2 positivity; like March to May.

Reviewer #3: The study is well designed and results are clear presented.

Concerning the results higher sensitivity for point of care devices was found, compared with other publications. The absence of cross-reactivity to seasonal coronavirus and influenza A is a remarkable result. Evolution of the virus and antibodies production turns serology assays a complex and hard work.

Although Authors did not consider strengths or weaknesses of the work.

To date the samples for serology PCR day was used, but it would be better to date from onset of symptoms.

The study is small and results probably will need to be reproduced in larger studies, in different places and in systematic reviews.

Line 268 – we chose – please correct.

6. PLOS authors have the option to publish the peer review history of their article (what does this mean?). If published, this will include your full peer review and any attached files.

Reviewer #1: No

Reviewer #2: No

Reviewer #3: No

---

## [Author Response · Author response to Decision Letter 0]

22 Dec 2020

Ethics changes had been requested previous to the decision letter and author stated the following:

The University of East Anglia Ethics Committee was consulted and confirmed no extra ethics was required as all samples were anonymized and this is covered by the UEA blanket Ethics for method development.

We also consulted the UK National Heath System Health Research Authority online tool kit which confirmed no supplemental Ethics or patient consent were required.

---

## [Decision Letter · Decision Letter 1]

11 Jan 2021

Performance of SARS-CoV-2 Serology tests: Are they good enough?

PONE-D-20-31021R1

Dear Dr. Piec,

We’re pleased to inform you that your manuscript has been judged scientifically suitable for publication and will be formally accepted for publication once it meets all outstanding technical requirements.

Kind regards,

Muhammad Adrish

Academic Editor

PLOS ONE

Additional Editor Comments (optional):

All comments have been addressed

Reviewers' comments:

Reviewer's Responses to Questions

**Comments to the Author**

1. If the authors have adequately addressed your comments raised in a previous round of review and you feel that this manuscript is now acceptable for publication, you may indicate that here to bypass the “Comments to the Author” section, enter your conflict of interest statement in the “Confidential to Editor” section, and submit your "Accept" recommendation.

Reviewer #1: All comments have been addressed

Reviewer #2: All comments have been addressed

Reviewer #3: All comments have been addressed

2. Is the manuscript technically sound, and do the data support the conclusions?

Reviewer #1: Yes

Reviewer #2: Yes

Reviewer #3: (No Response)

3. Has the statistical analysis been performed appropriately and rigorously? 

Reviewer #1: Yes

Reviewer #2: Yes

Reviewer #3: (No Response)

4. Have the authors made all data underlying the findings in their manuscript fully available?

Reviewer #1: Yes

Reviewer #2: Yes

Reviewer #3: (No Response)

5. Is the manuscript presented in an intelligible fashion and written in standard English?

Reviewer #1: Yes

Reviewer #2: Yes

Reviewer #3: (No Response)

6. Review Comments to the Author

Reviewer #1: Comment: The manuscript# PONE-D-20-31021R1 entitled “Performance of SARS-CoV-2

Serology tests: Are they good enough?” assessed the performance of SARS-CoV-2 serologic tests available in clinical practices. The topic is highly relevant in the current settings. In this revised manuscript authors have addressed the comments and suggestions provided by each of the reviewers.

Reviewer #2: Author addressed all the comments properly. It is a great opportunity for me to be a reviewer of this manuscript which is related to COVID-1.

Reviewer #3: (No Response)

7. PLOS authors have the option to publish the peer review history of their article (what does this mean?). If published, this will include your full peer review and any attached files.

Reviewer #1: No

Reviewer #2: No

Reviewer #3: No

---

## [Editor Report · Acceptance letter]

2 Feb 2021

PONE-D-20-31021R1 

Performance of SARS-CoV-2 Serology tests: Are they good enough? 

Dear Dr. Piec:

I'm pleased to inform you that your manuscript has been deemed suitable for publication in PLOS ONE. Congratulations! Your manuscript is now with our production department. 

Kind regards, 

on behalf of

Dr. Muhammad Adrish 

Academic Editor

PLOS ONE